

# Comments on black hole interiors and modular inclusions

**Ro Jefferson**

Max Planck Institute for Gravitational Physics (Albert Einstein Institute),
Am Mühlenberg 1, D-14476 Potsdam-Golm, Germany

rjefferson@aei.mpg.de

## Abstract

We show how the traversable wormhole induced by a double-trace deformation of the thermofield double state can be understood as a modular inclusion of the algebras of exterior operators. The effect of this deformation is the creation of a new region of spacetime deep in the bulk, corresponding to a non-trivial center between the left and right algebras. This set-up provides a precise framework for investigating how black hole interiors are encoded in the CFT. In particular, we use modular theory to demonstrate that state dependence is an inevitable feature of any attempt to represent operators behind the horizon. Building on this geometrical structure, we propose that modular inclusions may provide a more precise means of investigating the nascent relationship between entanglement and geometry in the context of the emergent spacetime paradigm.

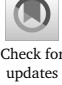
# 1 Introduction

One of the most fascinating connections to emerge from the study of holography is that between entanglement entropy and spacetime geometry. Evidence for this relationship made a dramatic first appearance in the Bekenstein-Hawking formula [1, 2], relating the entropy of a black hole to the surface area of its horizon. This naturally lead to the holographic bound [3], and the concomitant realization that information is stored holographically [4–6]. AdS/CFT [7] provides a concrete realization of this idea, in which the connection between entropy and area is encapsulated in the Ryu-Takayanagi formula [8–10]. Since then, entanglement has come to play a fundamental role in attempts to reconstruct the bulk from CFT data; see, e.g., [11–16].

Early efforts [16–19] to understand holographic entanglement entropy and its relation to geometry more deeply capitalized on the modular hamiltonian, which encodes certain thermodynamic properties of the vacuum state. Of course, the relationship between modular theory and thermodynamic properties of the vacuum is well-known in the AQFT literature (see, e.g., [20–24]), but only recently are physicists beginning to uncover the implications for entanglement and bulk reconstruction in the holographic context. In part, this difficulty stems from the fact that there are notoriously few cases in which an explicit expression for the modular hamiltonian is known.[1] The most famous example is that of Rindler space, in which the modular hamiltonian is proportional to the generator of Lorentz boosts [21, 22]:

$$H = 2\pi K = 2\pi \int_{x>0} \mathrm{d}^{d-1}x \, x \, T_{00}(x) \,. \tag{1}$$

Additionally, by conformally mapping the Rindler wedge to the domain of dependence of the spherical region $|x| < R$, one obtains the corresponding expression for a spherical entangling surface in a CFT:

$$H = 2\pi \int_{|x|<R} \mathrm{d}^{d-1}x \, \frac{R^2 - r^2}{2R} T_{00}(x) \,. \tag{2}$$

The Rindler result (1) has also been extended to more general regions bounded by null planes in a number of recent works, and exhibits a close connection to null energy conditions in QFT; see [25–29]. It also suggests an intimate relationship between the physics of horizons and that of entanglement. The holographic manifestation of this relationship is the Ryu-Takayanagi formula [8,9,30], relating the entanglement entropy of some boundary subregion with the area of the corresponding minimal surface in the bulk. Indeed, an underlying theme of the present note is the suggestion that the inherent structure of modular inclusions provides an alternative, and potentially deeper perspective on the emergent relationship between entanglement and geometry, to wit: *why* the Ryu-Takayanagi relation should be true.

In recent years, related ideas from modular theory have shown increasing utility in elucidating the interplay between entanglement, geometry, and locality [26, 27, 29, 31–44]. Such techniques are especially promising in attempts to reconstruct spacetime within holographic shadows [45], and behind the horizon of a black hole in particular. A notable example in this vein is the work of Papadodimas and Raju [46–50], who represented the black hole interior in terms of "mirror operators" in the exterior algebra. There has been some confusion in the literature over the *state-dependent* nature of these operators; that is, while they correctly represent the interior in any given state of the CFT, they do not exist as well-defined operators in themselves. One of the main objectives of this note is to demystify this state dependence—specifically, we shall demonstrate that state dependence is an inherent property of any attempt to represent physics behind horizons. Indeed, far from being a pathological feature, the necessity of state dependence can be understood from the imposition that such operators be both

---

[1]Strictly speaking, there are subtleties associated with attempting to define the modular hamiltonian for localized regions, but such technicalities of rigour will not be relevant here.

unitary and local (that is, localized to the exterior). As we will review in section 3, this can be understood from Tomita-Takesaki theory, and makes use of various properties of the modular operator acting on the vacuum state.

The set-up in which our investigations take place is the traversable wormhole recently constructed by Gao, Jafferis, and Wall [51]. Starting with the thermofield double (TFD) state (3) dual to an eternal black hole, they showed how a relevant double-trace deformation that couples the two CFTs allows a null observer to travel between the previously unbridgeable spacetimes. As we will explain in section 2, this is made possible due to the fact that the bulk exterior wedges are no longer disjoint, but instead overlap in a region corresponding to the creation of a non-trivial center shared by the left and right algebras of exterior operators. Our approach is inspired by algebraic quantum field theory (AQFT), and builds on the work of Papadodimas and Raju mentioned above.[2] In particular, we will demonstrate that the traversable wormhole induced by this process can be understood in terms of a modular inclusion, whereby the exterior algebra is enlarged to include information previously hidden behind the horizon of the black hole. On the bulk side, this appears to correspond to an increase in the entanglement between the two exteriors, which manifests in the emergence of a new region of spacetime deep in the bulk.

In this context, we propose that the structure of modular inclusions provides a concrete testbed for investigating how the black hole interior is encoded in the boundary, as well as a new perspective on the necessity of state dependence in representing operators behind the horizon. Furthermore, we suggest that this framework holds promise for investigating the idea of emergent spacetime in the sense of [53], and illustrate how modular inclusions may provide a more precise means of formulating the notion of "it from qubit" [54] or ER=EPR [55]. Our focus is on highlighting what we regard as interesting physical connections between various ideas in the literature, rather than lengthy calculations, in the hopes of stimulating further work in this exciting area.

The remainder of this paper is organized as follows. In section 2, we briefly review the double-trace deformation utilized by [51] to create a traversable wormhole, and show how this can be understood as a modular inclusion of the algebras of exterior operators. Then in section 3, we use the inherent structure of these inclusions to show how an operator behind the horizon can be represented in the exterior algebra. This provides a new perspective on the mirror operators of Papadodimas and Raju, and suggests that state dependence – that is, the inability to represent an interior operator locally in either CFT – is an inevitable property stemming from the fundamental structure of quantum field theories. We then bring together various ideas in the preceding sections – in particular the suggestive relationship between spacetime and entanglement – in section 4, and propose a more precise way of addressing the notion of emergent spacetime or "it from qubit" by iterating this modular inclusion procedure. We close in section 5 by remarking on some interesting connections to other ideas in the literature. In particular, we propose that modular theory provides the ontological basis for the success of quantum error correction (QEC) in holography. In the interest of making this paper self-contained, we have included a criminally brief overview of some relevant aspects of AQFT – in particular Tomita-Takesaki modular theory – in appendix A. Lastly, appendix B contains some additional remarks on the entropy associated with the inclusion structure as applied to the TFD.

---

[2]We note that mirror operators have appeared previously in the context of these double-trace deformations in [52].

## 2 Double-trace deformations and modular inclusions

We shall work with the eternal black hole in AdS, which is dual to the thermofield double (TFD) state [56]

$$|TFD\rangle = \frac{1}{\sqrt{Z_\beta}} \sum_i e^{-\beta E_i/2} |i\rangle_L |i\rangle_R \ . \tag{3}$$

Note that we have chosen the time on both the left (L) and right (R) boundaries to be zero. This describes two entangled CFTs which are joined in the bulk by a wormhole, i.e., an Einstein-Rosen bridge [55]. However, the exterior regions are causally disconnected for all time, so an observer who enters the future horizon inevitably hits the singularity before she can possibly reach the other side.

In order to create a traversable wormhole, [51] perturbed the TFD by a relevant double-trace deformation of the form

$$\delta S = \int dt \, d^{d-1}x \, h(t,x) \mathcal{O}_L(t,x) \mathcal{O}_R(t,x) \,, \tag{4}$$

where $\mathcal{O}_{L,R}$ is a scalar primary inserted in the left or right CFT, and $h$ controls the strength and spacetime support of the perturbation. Note that the deformation is explicitly chosen to be relevant, which requires that the dimension of $\mathcal{O}_{L,R}$ be less than $d/2$. Thus we choose the alternative boundary condition in the UV, giving $\mathcal{O}_{L,R}$ scaling dimension $\Delta_-$, and flow to the standard $\Delta_+$ condition in the IR; see, e.g., [57–60].

The deformation (4) is finely-tuned to effect a decrease in the energy of the TFD, which [51] showed explicitly to linear order in $h$ for the BTZ black hole. The bulk interpretation is therefore that $\mathcal{O}_{L,R}$ sources a negative-energy shockwave that reduces the mass of the black hole, thereby slightly decreasing the horizon area (see fig. 1). From the perspective of either the left or right CFT, the associated entanglement entropy between the boundaries therefore also appears to decrease (see appendix B). We shall discuss this in more detail in section 4 in the context of van Raamsdonk's proposal [53] relating entanglement entropy to spacetime in the interior. As we shall argue below, the ramifications of this statement are rather more subtle, due to the fact that in this case the algebras of operators on the left and right sides no longer commute in the overlapping region in the right panel of fig. 1.

Let us consider the physical effect of this deformation in more detail. As emphasized in [51], the wormhole is only open for a small amount of proper time, proportional to the strength of the perturbation $h$. This is sufficient for light-rays from one boundary – if emitted at sufficiently early times – to propagate through the event horizon to reach the opposite boundary in the far future. Such a null observer is illustrated in fig. 1 by the dashed line. Of course, in the unperturbed TFD, the observer is doomed to hit the singularity after entering the black hole from the left; but the double-trace deformation causes a backreaction on the geometry, which seems to miraculously extract her from the interior, allowing her to propagate safely to the right CFT. This effect on the null observer is typically phrased as a time-advance that shifts the trajectory along the shockwave, and is indicated in the left panel of fig. 1.

The right panel offers a more physical picture: the shockwaves inject negative energy into the black hole, which causes the future horizon to shrink. Thus, for example, the horizon of the right black hole is shifted slightly inwards, and similarly for the black hole on the left. This opens up a causeway through which the null observer can safely travel. Note that insofar as event horizons are global properties of the spacetime, the observer is never technically inside the black hole; rather, she falls through the horizon from the left, and immediately finds herself outside the black hole on the right—passage through the wormhole is instantaneous. This is not to say that the deformation does not allow one to extract information about the black hole interior, but that one must be careful in specifying precisely what such statements mean.

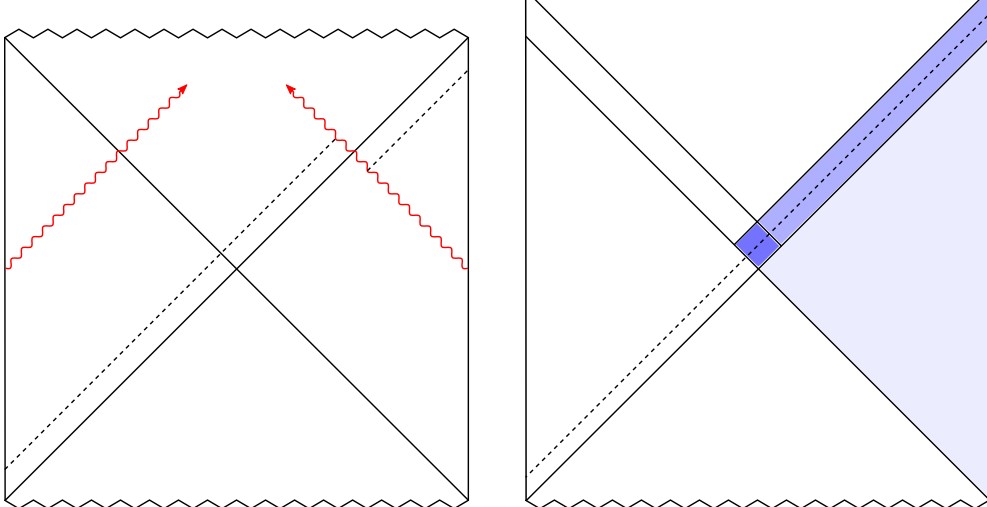

Figure 1: TFD under the double-trace deformation of [51]. (*Left*) The deformation manifests in the bulk as two negative-energy shockwaves (red wavy lines). A null observer (dashed line) who falls into the black hole from the left suffers a time-advance upon crossing the shockwave, which shifts her to the exterior region on the right. A more physical way of understanding this is that the negative-energy shocks lower the mass of the black hole, thereby causing the future horizon to shrink (*Right*). The result is that the exterior wedge increases in size (from light to dark blue), which opens up a causeway through which the observer can safely travel. Note that the left and right exterior regions are no longer independent, but overlap in the central diamond (darkest blue). The algebraic structure that corresponds to these nested wedges is a modular inclusion.

In the present case, we mean that the deformed CFT contains information about the region which would be causally inaccessible otherwise. The teleological nature of event horizons implies that the double-trace deformation does *not* allow information inside the black hole to escape in any local sense—that is, it does not extract an interior observer across the horizon, as this would violate causality. However, from the global perspective, the deformation does imply that information which would otherwise correspond to a region in the interior is encoded in the (enlarged) exterior algebra. This set-up thus provides a relatively well-controlled testbed for investigating the issue of how black hole interiors are encoded in the CFT. We propose that the structure of modular inclusions allows one to precisely formulate and address such questions.

The reader unfamiliar with Tomita-Takesaki modular theory may wish to consult appendix A before proceeding, as we shall rely on some basic notions from AQFT henceforth. We shall denote the von Neumann algebra[3] corresponding to the original left/right bulk wedge by $\mathcal{N}_{L,R}$, and the algebra of the larger wedge in the presence of the deformation by $\mathcal{M}_{L,R}$. Clearly, $\mathcal{N}_L \subset \mathcal{M}_L$ and $\mathcal{N}_R \subset \mathcal{M}_R$ (cf. fig. 1, in which $\mathcal{N}_R$ corresponds to the light blue wedge, and $\mathcal{M}_R$ extends this to include the dark blue regions). Note that to minimize notational excesses, we shall often conflate the algebra with the corresponding spacetime region that is its domain, but the distinction should be kept in mind. We shall also use the same notation to refer to the dual boundary algebra, and trust that the distinction, when relevant, will be clear from context. It is important to note that these algebras only contain exterior operators; namely, light CFT

---

[3]Strictly speaking, it is necessary to truncate the algebra of CFT operators in order to preserve the separating property of the vacuum state; see [47, 48] and further discussion below.

operators on the boundary, dual to exterior fields in the bulk (see [47] and further discussion below). One way to see this is to observe that in the bulk, the (full) modular operator – which generates an automorphism on the algebra – has no support in the black hole interior. This is easy to see in the case of Rindler space, and the modular flow in the present geometry respects the same structure. Consistency with entanglement wedge reconstruction then implies that the only (state-independent) operators in the boundary algebra likewise correspond to the exterior. This can also be understood from the requirement that the bulk and boundary modular flows agree [32], and is consistent with the algebraic truncation discussed below.

Lastly, we shall also suppress the subscript on $\mathcal{N}, \mathcal{M}$ when we have no need to specify the wedge; e.g., we may generally speak of the inclusion $\mathcal{N} \subset \mathcal{M}$. We take the Hilbert space $\mathcal{H}$ on which the algebras act to be equipped with the common cyclic and separating vector $\Omega$, and denote by $\Delta_{\mathcal{N}, \mathcal{M}}$ and $J_{\mathcal{N}, \mathcal{M}}$ the modular operator and conjugation, respectively. If the condition $\Delta_{\mathcal{M}}^{it} \mathcal{N} \Delta_{\mathcal{M}}^{-it} \subset \mathcal{N}$ is satisfied $\forall t \leq 0 \in \mathbb{R}$, then $\mathcal{N} \subset \mathcal{M}$ is called a *modular inclusion* [61].[4]

This structure was originally introduced by Wiesbrock [61], and is briefly summarized in appendix A. The basic idea is that the algebras $\mathcal{N}$ and $\mathcal{M}$ are related by a one-parameter unitary group $U(a) = e^{iap}$ acting on $\mathcal{H}$, where $a \in \mathbb{R}$ and the generator $p$ is defined as

$$p \equiv \frac{1}{2\pi}(\ln \Delta_{\mathcal{N}} - \ln \Delta_{\mathcal{M}}) \ . \tag{5}$$

Since $\mathcal{N} \subset \mathcal{M}$, $\Delta_{\mathcal{M}}^{1/2} \leq \Delta_{\mathcal{N}}^{1/2}$ in the sense of quadratic forms,[5] and hence $\langle p \rangle \geq 0$ with equality iff $\mathcal{M} = \mathcal{N}$. By expressing the modular operators in terms of the associated modular hamiltonians $K_{\mathcal{M}, \mathcal{N}}$,

$$\Delta_{\mathcal{M}, \mathcal{N}} = e^{-K_{\mathcal{M}, \mathcal{N}}} \ , \tag{6}$$

the above becomes

$$p = \frac{1}{2\pi}(K_{\mathcal{M}} - K_{\mathcal{N}}) \ . \tag{7}$$

In other words, $p$ is the difference in the generators of Lorentz boosts for the two wedge regions $\mathcal{M}$ and $\mathcal{N}$. More precisely, the Lie product formula allows us to express $U(a)$ as

$$U(a) = \lim_{n \to \infty} \left( \Delta_{\mathcal{M}}^{\frac{ia}{2\pi n}} \Delta_{\mathcal{N}}^{\frac{-ia}{2\pi n}} \right)^n = \lim_{n \to \infty} \left( e^{-iaK_{\mathcal{M}}/n} e^{iaK_{\mathcal{N}}/n} \right)^n \ , \tag{8}$$

and thus $U(a)$ can be thought of as a sequence of infinitesimal Lorentz transformations in $\mathcal{M}$ and $\mathcal{N}$. Note that since $\mathcal{N}$ is a subalgebra of $\mathcal{M}$, it is possible for $K_{\mathcal{M}}$ to be colinear with $K_{\mathcal{N}}$, in which case $[K_{\mathcal{M}}, K_{\mathcal{N}}] = 0$ and (8) reduces to the combined boost. In the present case, where $\mathcal{M}$ and $\mathcal{N}$ share a horizon, $p$ can be expressed as a simple translation along the null ray, cf. [29].

As mentioned in the introduction, the modular operator $\Delta$ is known explicitly in very few examples; accordingly, while it plays a central role in describing the structure of von Neumann algebras – particularly through the modular theory developed by Tomita and Takesaki – a general understanding of its physical significance is incomplete. However, it is a well-known result in algebraic quantum field theory that modular groups of wedge algebras act as Lorentz boosts [21] (see also [24, 62]). This allows one to write an explicit expression for the modular hamiltonian for Rindler wedges; in particular, it implies that $\Delta_{\mathcal{N}}^{it} \mathcal{N} \Delta_{\mathcal{N}}^{-it} \subset \mathcal{N}$ is the algebra of all (Lorentz-boosted) operators in the Rindler wedge corresponding to the algebra $\mathcal{N}$. Of course, the present AdS geometry differs from Rindler, but considering the modular flow in the latter is useful for the purpose of gaining some familiarity with this construction.

---

[4]Specifically, our sign convention here indicates a $(-)$-half-sided modular inclusion; see appendix A.

[5]That is, the operator relation $A \leq B$ means that their inner products satisfy $\langle Ax, x \rangle \leq \langle Bx, x \rangle$ for all elements $x \in \mathcal{H}$. The inverse, $A^{-1} \geq B^{-1}$, holds if $A$ and $B$ are positive-definite.

Intuitively, one is tempted to think of the condition $\Delta_{\mathcal{M}}^{it}\mathcal{N}\Delta_{\mathcal{M}}^{-it} \subset \mathcal{N}$ as imposing that Lorentz boosts with respect to the larger wedge $\mathcal{M}$ do not take one beyond $\mathcal{N}$.[6] And indeed, this intuition agrees if we consider nested Rindler wedges on the same spacelike Cauchy slice. In the present case however, $\mathcal{M}$ shares a null boundary with $\mathcal{N}$, so this naïve interpretation is not quite right; rather, the condition involves both a forward and backwards transformation, and the combination indeed keeps one in $\mathcal{N}$. In fact, the case in which the wedge algebras $\mathcal{N} \subset \mathcal{M}$ differ by a translation along the null direction corresponds to a *modular translation* [23]. Such null translations are a special instance of modular inclusions (see, e.g., [63–65]), but we shall continue to use the term "inclusion" for generality.

In this framework, the situation depicted in fig. 1 is that the algebra of the original right/left exterior $\mathcal{N}_{L,R}$ is enlarged – via the deformation (4) – to $\mathcal{M}_{L,R}$, which includes information previously encoded in the black hole interior.[7] A key feature of this construction is that in the presence of the double-trace deformation, the left and right wedges are no longer independent, but overlap in the darkest blue diamond in the center (fig. 1, right panel). That is, the algebras $\mathcal{M}_L$, $\mathcal{M}_R$ share a non-trivial center $\mathcal{C} = \mathcal{M}_L \cap \mathcal{M}_R$ with support in this overlapping double-cone, and consequently $[\mathcal{M}_L, \mathcal{M}_R] \neq 0$. The physical ramification of this is that information in the region of $\mathcal{M}$ behind the horizon does not admit a local representation in either CFT. We shall demonstrate this explicitly in section 3, which provides a new perspective on the state-dependent mirror operators of Papadodimas and Raju [48].

Before proceeding, let us clarify what we mean by saying that the exterior algebra is enlarged. A large-$N$ CFT with a local bulk dual exhibits a hierarchy between low- and high-dimension operators. Loosely speaking, heavy operators represent the interior microstates of the black hole, while light operators probe the exterior spacetime [66–70]. Thus, while the dimensionality of the total Hilbert space remains unchanged under the inclusion structure above, the demarcation between heavy and light operators is shifted in momentum space by an amount proportional to $h$ (i.e., the energy of the shock), such that some operators on the heavy side of the division are now considered light. Pictorially, one sees that an operator $\mathcal{O} \in \mathcal{M}$ just behind the horizon (the dark blue region in fig. 1) is an exterior operator with respect to $\mathcal{M}$, but corresponds to a state in the interior with respect to $\mathcal{N}$. This categorical change can be understood from the fact that the deformation (4) induces a flow along the RG (though still above the Hawking-Page transition), whereupon more low-energy operators are required to describe the greater exterior spacetime.[8] In the bulk, this is reflected in the decreased size of the eternal black hole, corresponding to the fact that temperature of each CFT has decreased; we will comment further on this below.

This division is also relevant if we wish to apply the theory of von Neumann algebras. A natural identification of the exterior algebra is the set spanned by all products of single-trace operators in the CFT. However, while this set certainly closes to form an algebra, it also contains operators which annihilate the vacuum state, and hence does not preserve the separating property on which modular theory depends.[9] Another way to say this is that allowing arbitrary products of single-trace operators eventually produces heavy states whose backreaction on the geometry can no longer be ignored. The necessity of truncating the exterior algebra implies

---

[6]We caution the reader again that we are superloading the notation for algebras and regions here.

[7]As emphasized above, event horizons are global properties of the space*time*, and hence terms like "original" and "previous" refer to the undeformed CFT rather than some temporally prior state. Said differently, the shift in the horizons does not correspond to any dynamical process in the bulk.

[8]More concretely: by the state-operator correspondence, operators are classified as heavy or light depending on whether the energy of the corresponding state is above or below some positive number (e.g., $c/12$ for $2d$ CFTs). In the high temperature (black hole) phase, the sum of energy eigenstates is inversely proportional to $\beta$ [66], so decreasing the temperature of a large black hole forces some of these heavy states below the cutoff.

[9]In particular, arbitrary products of operators would allow one to approximate the hamiltonian $H$ to any desired accuracy; and since $H|\Omega\rangle = 0$, it must be explicitly excluded. See [48] for further discussion.

that strictly speaking, Tomita-Takesaki theory does not apply. As discussed in [47,48] however, provided one limits the set of allowed observables to $\lesssim O(1)$ products of fundamental operators, the correlators obtained via the Tomita-Takesaki construction agree to leading order in the large-$N$ expansion with those obtained directly from the TFD. There is an interesting parallel here with the necessity of remaining in the code subspace in the context of quantum error correction; see the discussion in section 5. Hence, for the purposes of this note, we shall operate under the assumption that the algebraic approach can be meaningfully applied provided we do not examine very high-point correlators, i.e., that we limit inquiries to the domain of low-energy effective field theory, such that corrections appear only at higher-orders in $1/N$.

## 3  State-dependent interiors

In this section, we shall demonstrate how the inclusion structure introduced above can be used to clarify how interior operators are encoded in the boundary field theory. We shall first review the construction of mirror operators introduced by Papadodimas and Raju, following [50] (see also [46–49] for extensive discussion). These were originally arrived at following more physical arguments, and have led to some confusion/controversy as to the precise meaning and significance of "state dependence". In this respect, one of our aims is to show that this state dependence arises quite naturally from the algebraic perspective, and indeed is an inevitable consequence of attempts to represent operators outside their causal domain. In fact, far from being a pathological feature, this state dependence dovetails with the idea of bulk reconstruction as a quantum error correction scheme, as we shall remark in sec. 5.

Mirror operators create excitations behind the horizon, and therefore serve as probes of the black hole interior. Their construction can be neatly understood from Tomita-Takesaki theory – a basic introduction to which is given in appendix A – and relies on the fact that the conjugation operator $J$ induces an isomorphism between the von Neumann algebra $\mathcal{A}$ and its commutant $\mathcal{A}'$:

$$J\mathcal{A}J = \mathcal{A}' , \tag{9}$$

cf. (39). That is, $\forall \mathcal{O} \in \mathcal{A}$, we may define an element $J\mathcal{O}J \in \mathcal{A}'$ such that $[\mathcal{O}, J\mathcal{O}J] = 0$.

Explicitly, let $\mathcal{O} \in \mathcal{A}$ be a unitary operator with support in the right Rindler wedge $R$.[10] This creates a state $|\psi\rangle = \mathcal{O}|\Omega\rangle$ which is indistinguishable from the vacuum state for all observers in the left Rindler wedge $L$, i.e., all operators $\mathcal{O}' \in \mathcal{A}'$:

$$\langle\psi|\mathcal{O}'|\psi\rangle = \langle\Omega|\mathcal{O}^\dagger\mathcal{O}'\mathcal{O}|\Omega\rangle = \langle\Omega|\mathcal{O}'|\Omega\rangle , \tag{10}$$

where in the last step, we used the fact that $[\mathcal{O}, \mathcal{O}'] = 0$. Note that this wouldn't have worked if $\mathcal{O}$ were non-unitary. Now consider the state

$$|\phi\rangle = \Delta^{1/2}\mathcal{O}|\Omega\rangle . \tag{11}$$

Perhaps surprisingly, this state is indistinguishable from the vacuum for an observer in $R$, i.e., it represents an excitation localized entirely outside $R$![11] As demonstrated in [50], this is straightforward to show, using the properties of the modular conjugation $J$:

$$|\phi\rangle = J^2\Delta^{1/2}\mathcal{O}|\Omega\rangle = JS\mathcal{O}|\Omega\rangle = J\mathcal{O}^\dagger|\Omega\rangle = J\mathcal{O}^\dagger J|\Omega\rangle = \mathcal{O}'|\Omega\rangle , \tag{12}$$

---

[10]This framework holds generally, but we have chosen Rindler to make contact with the physical case of interest.

[11]As emphasized in [50], one cannot blithely say that the state is localized in $L$, since suitable time-evolution may propagate the excitation through the future Rindler horizon. For this reason, we've temporarily dropped our cavalier convention of conflating the algebra with its region of support.

where $\mathcal{O}' \equiv J\mathcal{O}^\dagger J$ is an operator in the commutant $\mathcal{A}'$. Indistinguishably of $|\phi\rangle$ from the vacuum for observers in $R$ then follows by the same logic as (10). However, while it is true that the state $|\phi\rangle$ is (initially) localized entirely in $L$, the same is *not* true for the operator $\Delta^{1/2}\mathcal{O}$! This is due to the fact that, as mentioned previously, $\Delta$ has support in both $L$ and $R$, and hence the operator $\Delta^{1/2}\mathcal{O}$ belongs to neither $\mathcal{A}$ nor $\mathcal{A}'$. This example highlights the subtle yet crucial distinction between localized states vs. localized operators. Another way to emphasize this is that, as operators,

$$\mathcal{O}' \neq \Delta^{1/2}\mathcal{O}, \tag{13}$$

since these exist in different algebras, even though as states,

$$\mathcal{O}'|\Omega\rangle = \Delta^{1/2}\mathcal{O}|\Omega\rangle \,, \tag{14}$$

as shown in (12).[12] In the context of firewalls [71,72] this structure allowed Papadodimas and Raju to construct a *mirror operator* $\tilde{\mathcal{O}}$ which represents an excitation localized behind the horizon from the perspective of an external observer. The issue of state dependence arises from the fact that, as we have just seen, $\tilde{\mathcal{O}}$ does not strictly exist as a well-defined operator in the exterior algebra, but does exhibit the correct expectation values when inserted in low-energy correlators. It is in this sense that the mirror operator serves as a probe of physics in the black hole interior.

The analogous statement in the presence of the traversable wormhole in fig. 1 is that one can map an interior state in $\mathcal{M}$ to an exterior state in $\mathcal{N}$, but one cannot do the same for operators. Physically, this means that the information in the interior region does not admit a local representation in either CFT. Of course, this was already implicit in the mirror construction above, but we shall now rephrase it explicitly using the structure of modular inclusions; subsequently, in section 4, we shall use this to strengthen the idea that the interior spacetime emerges from quantum entanglement [53]. Fundamentally, the lack of a local representation in either exterior stems from the fact that $\mathcal{M}_L$ and $\mathcal{M}_R$ have a non-trivial center $\mathcal{C}$. Hence our goal is to obtain an analogue of the mirror operator $\tilde{\mathcal{O}}$ above that represents the physics in the subregion of $\mathcal{M}_R$ that lies behind the horizon in the exterior algebra $\mathcal{N}_R$.

To that end, denote by $\mathcal{D}_R = \mathcal{M}_R - \mathcal{N}_R \subset \mathcal{M}_R$ the difference between the right wedges, that is, the region of $\mathcal{M}_R$ not contained in $\mathcal{N}_R$; see fig. 2. Since we shall work entirely from the perspective of the right algebras, we shall suppress the corresponding subscript on the operators in order to minimize clutter. Hence, let $D \in \mathcal{D}_R$ be a unitary operator in this region, which acts on the vacuum to create the state $|\psi\rangle \equiv D|\Omega\rangle$. Note that since the support of $\mathcal{D}_R$ lies entirely behind the horizon, this represents an excitation localized inside the black hole.[13] We shall now show that by using the unitary operator $U(a)$ and the modular conjugation $J$, we can find an equivalent state localized in the exterior algebra $\mathcal{N}_R$. The statement that $|\psi\rangle$ is localized to $\mathcal{N}_R$ means that the expectation value of any operator in the commutant, $\tilde{N}' \in \mathcal{N}_R'$, as well as any other operator $\tilde{D} \in \mathcal{D}_R$ (where $\tilde{D} \neq D$) is indistinguishable from its expectation value in the vacuum state, cf. (10). The first of these conditions is trivial, since $[D, \tilde{N}'] = 0$ by construction:[14]

$$\langle\psi|\tilde{N}'|\psi\rangle = \langle\Omega|D^\dagger \tilde{N}' D|\Omega\rangle = \langle\Omega|\tilde{N}'|\Omega\rangle \,. \tag{15}$$

But since $[D, \tilde{D}] \neq 0$, the second equality in the analogous computation for $\tilde{D}$ would fail. However, in the spirit of the mirror construction above, we can find a state which is equivalent to $|\psi\rangle$ for which this condition holds.

---

[12]I am grateful to Kyriakos Papadodimas for clarifying this distinction to me.

[13]In this section, the horizon is defined relative to the "original" exterior $\mathcal{N}_R$; obviously, the operators $D$ do not live in the interior relative to $\mathcal{M}_R$; cf. footnote 7.

[14]When discussing particular states, we imagine selecting a Cauchy slice that connects the left and right boundaries through the bulk, such that $D \in \mathcal{D}_R$ indeed commutes with any $N \in \mathcal{N}_R$ on the same slice.

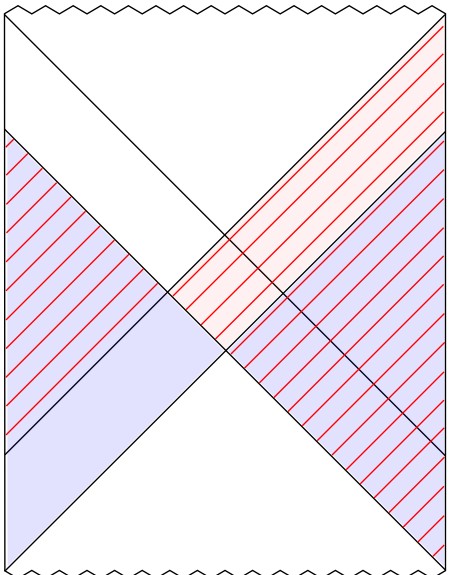

Figure 2: Same as the right panel of fig. 1, but with a greatly exaggerated shift in the horizon for illustration. $\mathcal{N}_R$ and $\mathcal{N}'_R$ are shaded blue, while $\mathcal{M}_R$ and $\mathcal{M}'_R$ are filled with red lines. Note that $\mathcal{N}'_R = \mathcal{N}_L$, but $\mathcal{M}'_R \neq \mathcal{M}_L$ since the inclusion breaks the left-right symmetry. The figure also illustrates the fact that $\mathcal{N}_R \subset \mathcal{M}_R \implies \mathcal{M}'_R \subset \mathcal{N}'_R$. The behind-the-horizon region $\mathcal{D}_R \subset \mathcal{M}_R$, i.e., the portion of $\mathcal{M}_R$ not contained in $\mathcal{N}_R$, is shaded red.

The tactic implemented in the following sequence of operations uses the structure of modular inclusions to map $D \in \mathcal{D}_R \subset \mathcal{M}_R$ into the commutant $\mathcal{M}'_R \subset \mathcal{N}'_R$, and then conjugate to an operator $N \in \mathcal{N}_R$. Of course, as operators, $N \neq D$, but the corresponding states will satisfy $N \ket{\Omega} = D \ket{\Omega}$. First, since $D \in \mathcal{M}_R$, there exists an operator $D' \in \mathcal{M}'_R$ such that $J_\mathcal{M} D' J_\mathcal{M} = D$. Hence

$$
\begin{aligned}
\ket{\psi} &= J_\mathcal{M} D' J_\mathcal{M} \ket{\Omega} = J_\mathcal{M} D' U(1) \ket{\Omega} = J_\mathcal{M} S U(-1) D'^\dagger \ket{\Omega} \\
&= J_\mathcal{M} S U(-1) D'^\dagger U(1) \ket{\Omega} = J_\mathcal{M} S N'^\dagger \ket{\Omega} = J_\mathcal{M} N' \ket{\Omega} \ ,
\end{aligned}
\tag{16}
$$

where we have used the fact that $J_\mathcal{M} \ket{\Omega} = U \ket{\Omega} = \ket{\Omega}$, and the existence of the operator $N'^\dagger \equiv U(-1) D'^\dagger U(1) \in \mathcal{N}'_R$ follows from (48). We can then trade $J_\mathcal{M}$ for conjugacy with respect to $\mathcal{N}$ (thereby enabling us to map $N'$ into $\mathcal{N}_R$) via (46), whence the above becomes

$$
J_\mathcal{N} U(2) N' \ket{\Omega} = J_\mathcal{N} S N'^\dagger U(-2) \ket{\Omega} = J_\mathcal{N} N' \ket{\Omega} = J_\mathcal{N} N' J_\mathcal{N} \ket{\Omega} = N \ket{\Omega} \ ,
\tag{17}
$$

where in the last step, we have defined $N \equiv J_\mathcal{N} N' J_\mathcal{N} \in \mathcal{N}_R$. We therefore have

$$
D \ket{\Omega} = N \ket{\Omega} \ ,
\tag{18}
$$

which achieves our goal of describing the excitation $\ket{\psi}$ entirely within the exterior wedge $\mathcal{N}_R$. We emphasize again that (18) holds only at the level of states, as we relied heavily on the invariance of the vacuum state (37) in deriving this relation. In particular, it is clear that as operators,

$$
D \neq N \ .
\tag{19}
$$

In fact, by tracing back through the operator identifications employed above, one sees that

$$
N = J_\mathcal{N} N' J_\mathcal{N} = J_\mathcal{N} U(-1) D' U(1) J_\mathcal{N} = J_\mathcal{N} U(-1) J_\mathcal{M} D J_\mathcal{M} U(1) J_\mathcal{N} = U(3) D U(-3) \ ,
\tag{20}
$$

where the last step follows from (46) in conjunction with (45); e.g.,

$$
J_\mathcal{N} U(-1) J_\mathcal{M} = J_\mathcal{M} U(-2) U(-1) J_\mathcal{M} = U(3) \ .
\tag{21}
$$

Thus, by creating the state $|\psi\rangle$ with $N$ in lieu of $D$, one can see that the aforementioned localization condition is satisfied,

$$\langle\psi|\,\tilde{D}\,|\psi\rangle = \langle\Omega|\,N^\dagger\tilde{D}N\,|\Omega\rangle = \langle\Omega|\,\tilde{D}\,|\Omega\rangle\;, \tag{22}$$

since $[N,\tilde{D}]=0$ for all $\tilde{D}\neq D\in\mathcal{D}_R$. Explicitly,

$$
\begin{aligned}
\langle\psi|\,\tilde{D}\,|\psi\rangle &= \langle\Omega|\,N^\dagger\tilde{D}N\,|\Omega\rangle = \langle\Omega|\,[U(3)DU(-3)]^\dagger\,\tilde{D}\,[U(3)DU(-3)]\,|\Omega\rangle \\
&= \langle\Omega|\,U(3)D^\dagger U(-3)\tilde{D}U(3)DU(-3)\,|\Omega\rangle \\
&= \langle\Omega|\,\big[U(1)D^\dagger U(-1)\big]\big[U(-2)\tilde{D}U(2)\big]\big[U(1)DU(-1)\big]\,|\Omega\rangle \\
&= \langle\Omega|\,\big[U(1)D^\dagger U(-1)\big]\big[U(1)DU(-1)\big]\big[U(-2)\tilde{D}U(2)\big]\,|\Omega\rangle = \langle\Omega|\,\tilde{D}\,|\Omega\rangle\;,
\end{aligned}
\tag{23}
$$

where in going to the last line, we used the fact that since $\mathcal{D}_R\subset\mathcal{M}_R$, (48) implies that $U(1)\mathcal{D}_R U(-1)\subset\mathcal{N}_R$, while $U(-a)\mathcal{D}_R U(a)\subset\mathcal{D}_R$ for $a>0$, and therefore these particular groupings of operators commute.

Of course, the state dependence inherent to (18) is essentially the same as that which underlies the Papadodimas-Raju construction of mirror operators discussed above. Furthering the analogy with these mirror operators, the state $N\,|\Omega\rangle$ contains information about the region behind the horizon, but the operator inequivalence (19) implies that this interior region is not probed by any (state-independent) local operator in either CFT. In this context, the necessity of state dependence [49] can be understood in at least two ways.

First, as emphasized above, the algebras $\mathcal{M}_L$ and $\mathcal{M}_R$ have non-trivial center $\mathcal{C}$, corresponding to the darkest blue double-cone in fig. 1. The physical consequence of this is that neither algebra is factor, and hence the algebra of bulk operators fails to respect the factorization of the boundary Hilbert space. Of course, even for the unperturbed TFD, it is an open question as to how the bulk spacetime emerges in a manner consistent with this factorization. It is a well-known albeit under-appreciated fact that the Hilbert space of even free quantum field theories (i.e., type III von Neumann algebras) does not factorize. There has been considerable interest in the factorization problem in the context of gauge [73–79] and gravitational [80–82] theories, but the failure in the case of free field theories is generally disregarded as the purview of some regularization procedure (see however [83], as well as [84] for a recent, physics-oriented review). Thus in the case of the algebras $\mathcal{N}_L$ and $\mathcal{N}_R$, one would typically attribute the failure of the bulk Hilbert space to factorize to UV divergences near the bifurcation horizon. In the case of $\mathcal{M}_L$ and $\mathcal{M}_R$ however, the problem is more severe due to the finite center. The lesson from the above is that in this case it is not possible to specify the complete bulk state behind the horizon using only state-independent operators on the boundary. One expects that a similar scenario holds in the case of a one-sided black hole in Minkowski space, namely that there does not exist a factorization $\mathcal{H}_I\otimes\mathcal{H}_E$ between the interior (I) and exterior (E) Hilbert spaces, which underlies many of the issues encountered in naïve models of black hole evaporation [85]; see section 5 for further discussion.

Another way to say this is that the inability to encode information about the black hole interior in terms of operators localized to either CFT is a natural consequence of the Reeh-Schlieder theorem [86]; specifically, that the existence of $N\in\mathcal{N}_R$ as an operator precludes unitarity thereof. This highlights an intriguing relationship between locality and unitarity embedded within the algebraic framework. While (18) gives the appearance of having created an excitation in $\mathcal{M}_R$ – specifically, in the region $\mathcal{D}_R\subset\mathcal{M}_R$ behind the horizon – by acting with an operator in $\mathcal{N}_R$, the preceding derivation implicitly relies on the modular operator $\Delta$, which lives in both the algebra and its commutant. To accomplish this feat with an operator whose support lies purely in $\mathcal{N}_R$ would require it to be non-unitary. This is nicely explained in the recent review by Witten [84], which we adapt to the present scenario as follows.

Suppose that $|\phi\rangle$ represents a state with an excitation behind the horizon, but within the domain of $\mathcal{M}_R$. For example, this could correspond to the action of a local bulk operator in the center $\mathcal{C} \subset \mathcal{M}_R$; cf. fig. 2. Following [84], we may define an operator $D \in \mathcal{D}_R \subset \mathcal{M}_R$ which has expectation value 1 in states which contain this excitation, and zero otherwise; i.e.,

$$\langle\phi|D|\phi\rangle = 1 \qquad \text{and} \qquad \langle\Omega|D|\Omega\rangle = 0 . \tag{24}$$

An intensely counterintuitive property of quantum field theories is that one can reconstruct this behind-the-horizon excitation arbitrarily well using operators which have no support in this region—in particular, an operator localized entirely in the exterior $\mathcal{N}_R$. This is the content of the Reeh-Schielder theorem, which asserts that the set of states in $\mathcal{N}_R$ is dense in $\mathcal{H}$.[15] Therefore, there exists an operator $N \in \mathcal{N}_R$ which approximates $|\phi\rangle$ arbitrarily well, and hence

$$\langle\phi|D|\phi\rangle \approx \langle\Omega|N^\dagger D N|\Omega\rangle = \langle\Omega|N^\dagger N D|\Omega\rangle , \tag{25}$$

where by definition $[N,D] = 0$. Now observe that if $N$ were unitary, (25) would imply a contradiction with (24). Thus, the guarantee on the existence of the local operator $N$ is furnished by Reeh-Schlieder at the cost of unitarity. At an operational level, this precludes the existence of any physical operation one might perform that effects such a gross violation of microcausality.

While the Reeh-Schlieder theorem and the particular consequences reviewed above are of course very well-known in the literature, extrapolating the above construction of modular inclusions demonstrates that information behind the horizon *cannot* be reconstructed by unitary operators in either CFT. This echoes claims in the literature, particularly [49] and related works, that there do not exist linear, state-independent operators in the CFT which describe the black hole interior. In this sense, the Reeh-Schlieder theorem may be regarded as a no-go result on the state-independent representation of any operator outside one's causal domain. Furthermore, in the case of the two-sided black hole dual to the TFD, our analysis suggests that physics in this interior region is encoded non-locally, in operators with support on both the left and right CFTs (that is, both $\mathcal{M}_L$ and $\mathcal{M}_R$). Another way to say this is that the spacetime region $\mathcal{C}$ that emerges as a result of the deformation (4) does so via the intrinsic entanglement of the vacuum $\Omega$. In this sense, the structure of modular inclusions provides a precise realization of the notion of emergent spacetime or "it from qubit" [53] (see also [54, 87]). We shall pursue this line of thought in the next section.

## 4 Building up spacetime with modular inclusions

In this section, we illustrate the possibility of formulating the notion of emergent spacetime using the framework of modular inclusions above. We first review the Shenker-Stanford prescription [88, 89] for creating long wormholes in the bulk by sending in positive-energy shockwaves. Geometrically, these have the opposite effect of the deformation discussed in sec. 2 insofar as they increase the size of the black hole. We argue that the structure of modular inclusions provides a unified way of treating both processes, and may thereby allow us to more rigorously investigate van Raamsdonk's proposal [53] that the interior spacetime is constructed from entanglement.

In [88, 89], Shenker and Stanford perturb the TFD by acting with unitary operators with energy $E \sim \beta^{-1}$ (that is, $O(1)$ in AdS units). This is much less than the mass of the black hole $M \sim G_N^{-1} \sim N^2$ in a large-$N$ gauge theory, and thus the immediate backreaction on the metric

---

[15]Physically, this is the statement that the vacuum (i.e., the cyclic and separating vector $\Omega$) is an infinitely entangled state—a fact which underlies the pervasive need for regulation whenever one restricts to a subspace.

is negligible. However, by inserting the source at some very early time $t_1 \ll 0$, the blueshift relative to the $t = 0$ slice which passes through the bifurcation surface implies that the energy is boosted to

$$E(t=0) \sim E(t_1)e^{2\pi t_1/\beta} , \qquad (26)$$

and hence backreaction becomes important when $t_1$ reaches of order the scrambling time, $t_* = \frac{\beta}{2\pi} \ln S_{\text{BH}}$. The result is that by taking $t_1$ much larger than $t_*$, we can create an arbitrarily high-energy shockwave in the bulk that shifts the horizon outwards. We shall refer to the shocks induced in this manner as SS-type, which one can visualize as injecting positive energy into the black hole. By repeating this process with alternating insertions in the far future and past,[16] one creates a long wormhole with a deep interior region causally disconnected from both CFTs. The first three steps in this procedure are illustrated in the top row of fig. 3, where we've introduced a pair of insertions,[17] indicated in blue, that shift the future horizon outwards to orange (first pair, left-most panel), then the past horizon to green (second pair, middle panel), and then the future horizon to pink (third pair, right-most panel). Letting $\mathcal{N}_0 \equiv \mathcal{N}_R$ denote the original right exterior wedge, and $\mathcal{N}_{-i}$ the wedge for the $i^{\text{th}}$ shock, one sees that the resulting structure of modular inclusions is

$$\ldots \subset \mathcal{N}_{-3} \subset \mathcal{N}_{-2} \subset \mathcal{N}_{-1} \subset \mathcal{N}_0 . \qquad (27)$$

In the limit of infinitely many shocks, correlations between the left and right boundaries vanish, and the geometry pinches off into two disconnected CFTs and their associated exterior regions [53, 89].

In contrast, the deformations employed by Gao, Jafferis, and Wall [51] that make the wormhole traversable do so by shrinking the future horizon. As elaborated above, one can think of these as negative-energy shockwaves, which we shall refer to as GJW-type. One can then imagine iterating this process by a sequence of relevant deformations, such that the duration of proper time for which the wormhole remains open increases. As illustrated in the second row of fig. 3, this effects precisely the reverse of the SS-type shockwaves above, in that it iteratively shrinks the interior region. Here, the GJW-type shocks are shown in red, and move the horizons inwards to orange (left), then green (middle), then pink (right). Letting $\mathcal{N}_i$ denote the wedge for the $i^{\text{th}}$ such insertion, with $\mathcal{N}_0$ the original wedge as before, we have

$$\mathcal{N}_0 \subset \mathcal{N}_1 \subset \mathcal{N}_2 \subset \mathcal{N}_3 \subset \ldots . \qquad (28)$$

In the language of enlarging the exterior algebra in section 2, this procedure sequentially shifts the cutoff between heavy and light operators in favour of the latter; eventually, in the limit of infinitely many shocks, the entire interior algebra has been subsumed by the exterior, and there are no more heavy states remaining. At this point the black hole has undergone a sort of forced evaporation, corresponding to the fact that the field theory has flown all the way to the Hawking-Page transition at the self-dual temperature $\beta \simeq \pi$. Formally, the final state would then be a single algebra of exterior operators which maximally fails to respect the Hilbert space factorization of the CFT.

As the illustration suggests, the SS- and GJW-type deformations can be viewed as two sides of the same coin, obtained by running the algebraic construction in opposite directions relative to the unperturbed initial state (3) at $t = 0$; cf. the two sequences (27) and (28). In this sense, the structure of modular inclusions offers a precise realization of the emergent

---

[16]This can be described in the field theory via a timefold [90].

[17]Note that in contrast to the deformations (4), these are independent and do not couple the two CFTs. Had we wished, we could have replaced all shocks on the left boundary with shocks propagating from the opposite direction on the right, but we have grouped them this way to facilitate the comparison with the GJW-type shocks below.

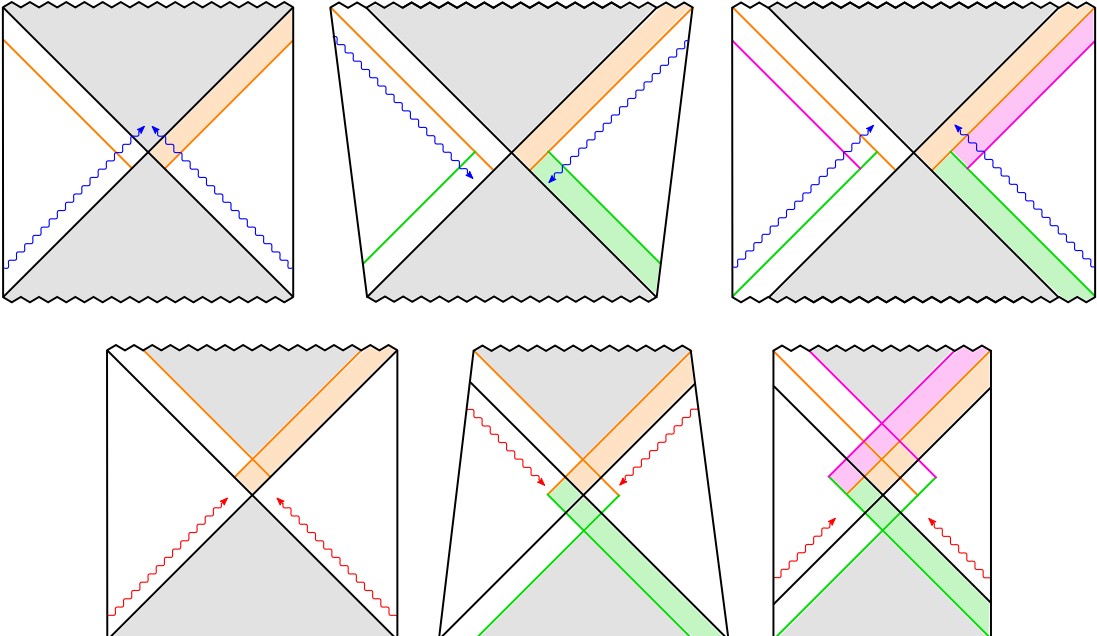

Figure 3: (*Top row*) TFD under sequential SS-type shocks, which move the horizons outwards. The original horizons are indicated in black, while their positions under the first three shocks are shown in orange (first shock, left), green (second shock, middle), and pink (third shock, right). The portion of the right exterior wedge $\mathcal{N}_R$ lost behind the horizon after each inclusion is shaded according to this same scheme, while the original interior is shaded grey. (*Bottom row*) TFD under sequential GJW-type shocks, using the same color scheme above. In this case, the horizons move inwards, so the interior of the black hole shrinks. Note the increasing overlap between the exterior wedges $\mathcal{N}_i$.

spacetime picture proposed by van Raamsdonk in [53]. He argued that decreasing the mutual information between the left and right CFTs in the TFD (3) corresponds to increasing the proper distance through the wormhole—and hence, running this reasoning in reverse, that the classical spacetime in the interior emerges from quantum entanglement between the two asymptotic boundaries. This was largely based on the observation that the mutual information imposes an upper bound on the correlators [91],

$$I(L,R) \geq \frac{(\langle \mathcal{O}_L \mathcal{O}_R \rangle - \langle \mathcal{O}_L \rangle \langle \mathcal{O}_R \rangle)^2}{2|\mathcal{O}_L|^2|\mathcal{O}_R|^2} \,, \tag{29}$$

in conjunction with the fact that certain two-point correlators of local operators decrease with the geodesic distance between them.

Instead of pinching-off the wormhole as in van Raamsdonk's story, let us consider the opposite direction, starting from two disentangled copies of the CFT. As explained above, we may think of this as an infinitely long wormhole, obtained by a large number of SS-type shocks. Starting in the upper right-most panel of fig. 3 and proceeding counter-clockwise, one sees that backing away from this limit corresponds to a sequence (27) of modular inclusions that slowly stitch together the interior spacetime. Eventually, at $t=0$, one recovers the TFD dual to the eternal black hole, whose interior is encoded in the entanglement between the two boundaries. If we continue past this point along the sequence (28), this interpretation suggests that we introduce so much entanglement that the previously independent bulk duals begin to merge into one; that is, one expects an increase in the mutual information corresponding

to the creation of the new exterior spacetime region $\mathcal{C}$ in the deep IR.[18] Schematically, the sequence of modular inclusions terminates at the self-dual temperature, when the black hole interior has been completely transferred to what is now a single exterior algebra.[19] Note that one can arrange for the geometry throughout this process to be smooth simply by taking each perturbation to be sufficiently weak (i.e., $h \ll 1$, which corresponds to $\frac{E}{4M}e^{2\pi t_i/\beta} \ll 1$ in [89]).

Of course, the heuristic picture above assumes that the algebraic approach can continue to be meaningfully applied throughout the entire process. As mentioned in section 2 however, Tomita-Takesaki theory is predicated on the existence of a cyclic and separating vector $\Omega$, and hence it was necessary to truncate the algebras to $\lesssim O(1)$ products of fundamental operators. This is exponentially less than the number of operators in the interior,[20] and therefore one might expect this construction to break down long before we manage to reconstruct an appreciable portion of the interior spacetime. However, recall that we create more room for formerly-heavy operators in the exterior as we flow into the IR. Thus one might hope that the necessary truncation is pushed farther into the UV relative to each new inclusion, but whether this can be self-consistently arranged requires a more detailed analysis. In particular, one might allow sufficient time between successive perturbations in the inclusion sequence (28) for the state to equilibrate at the new energy scale. Algebraically, this amounts to the requirement that we base the Tomita-Takesaki construction for each algebra $\mathcal{N}_i$ on the unique cyclic and separating vector $\Omega_i$ induced by the GNS representation. But then, while the non-uniqueness of the vacuum in this regard is not itself an issue, the ontological map between inequivalent representations is notoriously subtle; see, e.g., [93].

# 5 Discussion

In the preceding sections, we highlighted various ways in which the structure of modular inclusions may be used to shed light on black hole interiors, as well as on the relationship between entanglement, spacetime geometry, and the locality of information (or lack thereof) in holographic theories. We wish to close by discussing some connections to other ideas in the literature, and speculate on some interesting directions for further work in this vein.

In [94], it was proposed that AdS/CFT functions like a quantum error correction (QEC) scheme; see also [95–97] for applications of QEC in the context of firewalls. As alluded in section 2, the necessity of truncating the operator algebras is precisely analogous to the requirement, in this context, that one remain in the code subspace: both stem from the constraint that the backreaction on the metric remain sufficiently small. In the algebraic language, allowing arbitrary products of operators destroys the separating property of the vector $\Omega$ (e.g., by approximating the hamiltonian). Equivalently, one says that backreaction moves one beyond the code subspace, which implies a loss of the QEC code's capacity to correct erasures. Additionally, the authors of [94] put forth a number of simple examples to demonstrate that the bulk algebra cannot hold at the level of operators in the CFT. Rather, contradictions can be avoided by formulating the operators in terms of their action on the code subspace of states. This is precisely the same issue of state dependence we encountered above, couched in terms of quantum information theory. However, despite its epistemic utility, phrasing matters at such

---

[18]In fact, it is important to distinguish between bulk and boundary mutual information; see the discussion in appendix B.

[19]It is unclear whether this limit has a sensible holographic interpretation; at the very least, the existence of two distinct asymptotic regions in the bulk suggests that we must exercise more care in identifying the spacetime region of support. I am grateful to Raphael Bousso for raising this point.

[20]In order to reproduce the Hawking-Page phase structure, the spectrum of light operators must be sparse [66, 67,92]. Therefore heavy states dominate the microcanonical ensemble for the eternal black hole, and the density of states satisfies the Cardy formula, $\rho(E) \sim e^{S_{BH}(E)} \sim e^N$.

an operational level does not explain, ontologically, *why* holography should function as a QEC code. Modular theory thus provides the underlying physical reason for the efficacy of QEC in attempts to reconstruct both the bulk and the black hole interior. We hope that this approach may lend further insight into the fascinating connections between quantum information theory and high energy physics uncovered in recent years.

There is, however, an important technical distinction between the aforementioned applications of QEC and the algebraic approach discussed here, namely that the code subspace is typically regarded as tensor factor of the global Hilbert space; but as we have remarked above, the Hilbert space of continuum quantum field theory does not factorize.[21] This highlights an important connection between the problem of Hilbert space factorization and emergent spacetime. By construction, the TFD (3) – that is, the *boundary* Hilbert space – factorizes into a tensor product of the left and right CFTs, and therefore the bulk spacetime must emergent in a manner consistent with this factorization; see [98] for an interesting analysis in this context. This is especially pertinent in the context of the emergent spacetime paradigm discussed in section 4 above, in which gravity is conjectured to emerge from quantum entanglement. In particular, implicit in the definition of the von Neumann entropy, $S_A = -\text{tr}\rho_A \ln \rho_A$, is the assumption that the Hilbert space admits a tensor product decomposition $\mathcal{H} = \mathcal{H}_A \otimes \mathcal{H}_{\bar{A}}$, such that the subsystem $\rho_A$ is obtained by tracing out degrees of freedom in the complementary factor. But such a factorization fails already at the level of free field theory, to say nothing of gravity, and thus any attempt to found the latter on this premise is inherently contradictory! In light of the deepening connections between entanglement and geometry uncovered in recent years, it therefore seems well-worth investigating whether ideas from AQFT, such as the approach from modular theory advocated here, may provide a less problematic means of addressing such questions.[22]

As an aside, we remark that the above is in agreement with the Weak – as opposed to Strong – interpretation of black hole complementarity (BHC) [100, 101]. Strong BHC assumes independent Hilbert spaces for the interior and exterior; but this is problematic since, in addition to the factorization issue above, it would require consistent matching conditions in order to stitch the spacetime together along the entire event horizon. As noted in [101], it is also undesirable insofar as it makes the Hilbert space structure subordinate to the causal structure. This is the opposite of the standard formulation of QFT, in which microcausality naturally emerges from quantum mechanics in conjunction with special relativity and the clustering property [102]. Obviously, one would like to preserve this notion of emergent locality in any attempt to derive spacetime from entanglement. In contrast, Weak BHC posits a global Hilbert space, except that spacelike operators in the interior and exterior no longer commute. Naïvely, it appears that locality is badly broken; but the issue is again the same as that which underlies the need for state dependence, or the use of QEC in bulk reconstruction, namely that the interior cannot be represented in terms of state-independent operators in the exterior. Indeed, the harmony of the Weak interpretation with holography was noted already in [101] (see also [48]); we have merely strengthened the connection between these ideas.

As we emphasized in section 3, one can view state dependence – i.e., the inability to represent the interior of the black hole in terms of localized operators in the exterior algebra – as an inevitable property of quantum field theories that ultimately follows from the Reeh-Schlieder theorem. In our discussion of mirror operators, we demonstrated explicitly that this intrinsic non-locality is in fact necessary in order to preserve unitarity of the theory. Extrapolating this to the limiting procedure in which we built-up the bulk spacetime from modular inclusions in

---

[21]Technically, the issue is not whether the Hilbert space factorizes (e.g., there is no obstruction for separable states), but whether the algebra of operators respects this factorization; it is the latter which fails for $\mathcal{M}_L$ and $\mathcal{M}_R$ above. I am grateful to Chris Fewster for clarifying this distinction.

[22]See, e.g., [99] for a more algebraic treatment of entanglement entropy.

section 4 therefore contributes to the growing body of evidence that holography must be fundamentally non-local. Furthermore, this construction shows that, far from being a pathological quirk of existing bulk reconstruction schemes, non-locality and unitarity are inseparable sides of the same coin.

Our prescription for building-up spacetime via modular inclusions in section 4 was quite abstract; for example, our arguments from entanglement entropy and overlapping algebras do not suffice to show that one recovers, say, the bulk Einstein equations from this algebraic structure. However, this is precisely what was done in [103] for small perturbations of the vacuum by utilizing the first law of entanglement entropy (see, e.g., [34]), and Faulkner has shown [17] that the linearized Einstein equations arise quite naturally via the entropy-area relation implicit in this law. Indeed, as alluded above, the modular hamiltonian appears to encode aspects of the geometry in the thermodynamic properties of the vacuum. For example, the Unruh effect [104, 105] arises as a consequence of the fact that the vacuum is a KMS state with respect to the generator of Lorentz boosts, i.e., the modular hamiltonian [21, 22]. More generally, the modular operator arises as a natural distance measure used in defining the relative entropy between states (see, e.g., [106–109]), and connections between modular theory and the Poincaré group are very well-known in the AQFT literature [21, 24, 63–65, 110, 111]. Thus, while the proposal in section 4 is little more than a rough sketch, there are certainly grounds for suspecting that the structure of modular inclusions may enable one to make the nascent connection between entanglement and spacetime geometry much more solid. In doing so, it may shed light on the deeper reasons for the validity of the Ryu-Takayanagi formula and its generalizations, as well as provide a precise realization of the ER=EPR proposal [55] (see also related discussion in [49]). It would be very interesting to explore this more concretely, e.g., in conjunction with the behaviour of the dual field theory under the RG flow induced by (4).

Finally, it has been suggested that when considering dynamical information, entanglement entropy alone is insufficient to reconstruct the bulk spacetime in the wormhole's interior [112].[23] In particular, Susskind has proposed "holographic complexity" as the field-theoretic quantity that encodes the evolution of the Einstein-Rosen bridge, whose bulk dual is given by either the volume of the codimension-1 slice that connects the two CFTs [114], or the Einstein-Hilbert action of the Wheeler-DeWitt patch defined by null rays emitted at times $t_L$ and $t_R$ [115, 116]. While our understanding of complexity in field theory is still in its nascent stages (see initial works [117, 118] and subsequent extensions, including [119] for an application of circuit complexity to the TFD, and [120] for considerations in the context of double-trace deformations), preliminary evidence indicates that complexity does indeed probe global information to which entanglement entropy is insensitive [121]. Additionally, as one observes in fig. 3, the double-cone prescribed by the overlapping wedge regions is reminiscent of the Wheeler-deWitt patch behind the horizon; see in particular [122]. Hence, in light of the central role of the modular operator in information theory alluded above, such an approach may provide insight as to a more natural definition of complexity for general quantum field theories.

# Acknowledgements

It is a pleasure to thank Raphael Bousso, Diptarka Das, Chris Fewster, Ben Freivogel, Stefan Hollands, Nirmalya Kajuri, and Gabriel Wong for helpful discussions. I am especially grateful to

---

[23]In fact, holographic shadows [45] already pose barriers for existing reconstruction schemes, including those based on entanglement entropy. In the two-sided case [113] for example, the switchover effect prevents the entanglement surface from probing the interior geometry at late times.

Kyriakos Papadodimas for comments on an early draft of this manuscript, and for several stimulating discussions about the algebraic approach to reconstructing interiors. I would also like to thank Erik Verlinde for an engaging discussion about the application of modular inclusions in this context. Last but not least, I am indebted to Michal Heller for several helpful conversations and comments on an early draft, as well as for support and encouragement throughout the completion of this project. The Gravity, Quantum Fields and Information group at AEI is generously supported by the Alexander von Humboldt Foundation and the Federal Ministry for Education and Research through the Sofja Kovalevskaja Award.

# A  Tomita-Takesaki in a nutshell

In this appendix, we collect some results from AQFT, in particular the modular theory of Tomita and Takesaki, that we use in the main text. Our goal is to convey the basic ingredients as concisely as possible, and no pretense is made at mathematical rigour. The reader interested in a more mathematical treatment is referred to the classic book by Haag [20], or the excellent review by Halvorson [123], both of which are noteworthy for their emphasis on foundational issues. A more physics-oriented introduction is provided by the recent and pedagogically exemplary review by Witten [84], which is explicitly geared towards understanding entanglement in quantum field theory.

We begin with the Hilbert space $\mathcal{H}$ of some quantum field theory,[24] and denote by $\mathcal{B}(\mathcal{H})$ the set of all bounded linear operators acting on $\mathcal{H}$. A $C^*$-algebra $\mathcal{A}_\mathcal{U}$ is a subalgebra of $\mathcal{B}(\mathcal{H})$ consisting of all observables in the spacetime region $\mathcal{U}$, which is closed both under the norm topology and the adjoint operation. Note that the $C^*$-algebra does not contain unbounded operators. These are of central importance in physics, and include, e.g., the total energy, the number operator, and even the basic position and momentum observables in field theory. This motivates us to consider the weak closure of $\mathcal{A}_\mathcal{U}$, which defines a von Neumann algebra. Recycling notation, and suppressing the spacetime region $\mathcal{U}$, we henceforth refer to this von Neumann algebra as $\mathcal{A}$, and denote the commutant $\mathcal{A}'$, where

$$\mathcal{A}' = \{B \in \mathcal{B}(\mathcal{H}) : [A, B] = 0 \ \ \forall A \in \mathcal{A}\} \,. \tag{30}$$

By von Neumann's double commutant theorem, $(\mathcal{A}')' = \mathcal{A}'' = \mathcal{A}$. Note that we are implicitly assuming Haag duality, $\mathcal{A}'_\mathcal{U} = \mathcal{A}_{\mathcal{U}'}$, which applies when $\mathcal{U}$ and $\mathcal{U}'$ are causal complements.

We take $\mathcal{A}$ to be equipped with the cyclic and separating vector $\Omega$. *Cyclic* means that the states spanned by $\mathcal{O} \in \mathcal{A}$ are dense in $\mathcal{H}$. In other words, we can approximate any state in the total Hilbert space by acting on $|\Omega\rangle$ with operators whose support is limited to the subregion $\mathcal{U}$ over which $\mathcal{A}$ is defined (cf. Reeh-Schlieder). *Separating* means that $\mathcal{O}|\Omega\rangle = 0$ iff $\mathcal{O} = 0$. The vector $\Omega$ thus furnishes a representation of the vacuum state.

Given a von Neumann algebra $\mathcal{A}$, Tomita-Takesaki theory admits a canonical construction of the commutant $\mathcal{A}'$. The starting point is the antilinear map $S : \mathcal{H} \to \mathcal{H}$, defined by

$$S\mathcal{O}|\Omega\rangle = \mathcal{O}^\dagger|\Omega\rangle \,, \tag{31}$$

where recall that an antilinear map $f : V \to W$ between complex vector spaces $V, W$ satisfies $f(aX + bY) = a^* f(X) + b^* f(Y) \ \ \forall a, b \in \mathbb{C}$ and $X, Y \in V$. Note that $S$ is a *state-dependent operator*, insofar as it is defined in terms of its action on the cyclic and separating state $|\Omega\rangle$. We emphasize that there is nothing pathological about this state dependence; rather, it is an intrinsic feature of the algebraic construction.

---

[24]We are concerned with type III von Neumann algebras only.

For example, that the fact that $\Omega$ is separating is necessary for $S$ to be well-defined, otherwise there would exist an annihilation operator $a \in \mathcal{A}$ such that $a|\Omega\rangle = 0$, but $Sa|\Omega\rangle = a^\dagger|\Omega\rangle \neq 0$, which contradicts the definition (31). The fact that $\Omega$ is cyclic then ensures that it maps to a dense set of states on $\mathcal{H}$. Additionally, it follows from (31) that, as an operator, we must have $S^2 = 1$, and therefore $S$ is its own inverse, $S^{-1} = S$. As shown in [84], taking the hermitian conjugate yields the corresponding operator on the commutant, $S^\dagger = S'$. Lastly, and quite crucially, observe that taking $\mathcal{O} = \mathbb{1}$ in (31) implies that $S$ leaves the vacuum state invariant, $S|\Omega\rangle = |\Omega\rangle$.

Now, since $S$ is invertible, it admits a unique polar decomposition,

$$S = J\Delta^{1/2} , \tag{32}$$

where $J$ is an antiunitary operator (i.e., $\langle JA, JB\rangle = \langle A, B\rangle^\dagger \;\; \forall A, B \in \mathcal{H}$) called the *modular conjugation*, and $\Delta$ is a self-adjoint operator called the *modular operator*. The decomposition (32), combined with the fact that $\Delta^\dagger = \Delta$, implies that

$$\Delta = S^\dagger S , \tag{33}$$

which is sometimes taken as its definition. Additionally, since $\Delta$ is a positive hermitian operator, we may define the *modular hamiltonian*

$$K \equiv -\log(S^\dagger S) , \tag{34}$$

such that[25]

$$\Delta = e^{-K} . \tag{35}$$

Since both $S$ and $S^\dagger$ leave the vacuum state invariant, this property extends via (33) to the modular operator as well, $\Delta|\Omega\rangle = |\Omega\rangle$. Additionally, it is of central importance to note that $\Delta$ does *not* have support purely within the region $\mathcal{U}$, but acts on the full Hilbert space defined over both $\mathcal{A}$ and its commutant $\mathcal{A}'$.[26]

By way of final preliminaries, we note the following useful identities:

$$J\Delta^{1/2} = \Delta^{-1/2}J \quad \text{and} \quad J^2 = 1 \iff J^{-1} = J . \tag{36}$$

These essentially follow from the fact that $S^2 = 1$ in conjunction with the polar decomposition; see [84] for more details. Combining these with the aforementioned fact that $S' = S^\dagger$ then leads to $J' = J$ and $\Delta' = \Delta^{-1}$. Lastly, expressing $J$ in terms of $S$ and $\Delta$ via (32) reveals that it likewise leaves the vacuum invariant; i.e., we have

$$S|\Omega\rangle = J|\Omega\rangle = \Delta|\Omega\rangle = |\Omega\rangle . \tag{37}$$

This invariance of the vacuum features crucially in our analysis in section 3, as well as in the state-dependent mirror operators introduced by Papadodimas and Raju.

Now, the fundamental result of Tomita-Takesaki theory is comprised of the following two facts: first, the modular operator $\Delta$ defines a one-parameter family of *modular automorphisms*

$$\Delta^{it}\mathcal{A}\Delta^{-it} = \mathcal{A} , \quad \forall t \in \mathbb{R} . \tag{38}$$

---

[25]Note that $K$ is occasionally defined (e.g., in [84]) with an additional factor of $2\pi$, so that it precisely equals the generator of Lorentz boosts (cf. $H$ in the Rindler example (1)), but we prefer to keep these *a priori* distinct, following [50].

[26]The attentive reader may be disturbed by the implication that $K|\Omega\rangle = 0$, but this does not contradict the separating property of $\Omega$ since $K$ is not localized to any subregion. This, incidentally, is also why there are no truly local modular hamiltonians.

In other words, the algebra $\mathcal{A}$ is invariant under *modular flow*. Second, the modular conjugation induces an isomorphism between the algebra and its commutant $\mathcal{A}'$,

$$J\mathcal{A}J = \mathcal{A}' \,, \tag{39}$$

i.e., $\forall \mathcal{O} \in \mathcal{A}$, $J$ defines an element $\mathcal{O}' \equiv J\mathcal{O}J$ such that $[\mathcal{O}, \mathcal{O}'] = 0$. The isomorphism (39) is rather remarkable. For example, it allows one to map operators between the left and right Rindler wedges [50]—or, as shown in section 3, across the horizon of a black hole.

Motivated by the desire to understand more deeply the physical interpretation of these modular objects, Wiesbrock [61] introduced the notion of a *half-sided modular inclusion*.[27] Given the preliminaries above, we shall simply state the results from his paper that we employ in the main text. Further discussion and intuition for this structure is included in section 2. Modular inclusions are also treated in more detail in [23, 63, 64, 125], and have appeared recently in the field theory literature in [29].

Let $\mathcal{N} \subset \mathcal{M}$ be von Neumann algebras acting on $\mathcal{H}$, equipped with a common cyclic and separating vector $\Omega$. The associated modular operators and modular conjugations are respectively denoted $\Delta_{\mathcal{N}}$, $\Delta_{\mathcal{M}}$ and $J_{\mathcal{N}}, J_{\mathcal{M}}$. Further suppose that $\mathcal{N}$ is preserved under the modular flow induced by $\mathcal{M}$, specifically

$$\Delta_{\mathcal{M}}^{it}\mathcal{N}\Delta_{\mathcal{M}}^{-it} \subset \mathcal{N} \qquad \forall \mp t \leq 0 \in \mathbb{R} \,. \tag{40}$$

In this case, the inclusion $\mathcal{N} \subset \mathcal{M}$ is called $(\mp)$-half-sided modular. Note that allowing $t \in \mathbb{R}$ implies only the trivial solution $\mathcal{N} = \mathcal{M}$, hence the restricted range [125]. The algebras are related by a one-parameter unitary group

$$U(a) = e^{iap} \,, \quad a \in \mathbb{R} \,, \tag{41}$$

where the generator is defined as

$$p \equiv \frac{1}{2\pi}(\ln \Delta_{\mathcal{N}} - \ln \Delta_{\mathcal{M}}) \geq 0 \,, \tag{42}$$

cf. (5). We note that

$$U(t)\mathcal{M}U(-t) \subset \mathcal{M} \qquad \forall \pm t \geq 0 \,, \tag{43}$$

i.e., $(\mp)$-half-sided inclusions are associated with $(\pm)$-half-sided translations; see [125] for more details. With this structure in hand, one has the following relations [61]:

$$\Delta_{\mathcal{M}}^{it}U(a)\Delta_{\mathcal{M}}^{-it} = \Delta_{\mathcal{N}}^{it}U(a)\Delta_{\mathcal{N}}^{-it} = U\big(e^{\mp 2\pi t}a\big) \,, \qquad \forall t, a \in \mathbb{R} \tag{44}$$

$$J_{\mathcal{M}}U(a)J_{\mathcal{M}} = J_{\mathcal{N}}U(a)J_{\mathcal{N}} = U(-a) \,, \qquad \forall a \in \mathbb{R} \tag{45}$$

$$J_{\mathcal{N}}J_{\mathcal{M}} = U(2) \,, \tag{46}$$

$$\Delta_{\mathcal{N}}^{it} = U(1)\Delta_{\mathcal{M}}^{it}U(-1) \,, \qquad \forall t \in \mathbb{R} \tag{47}$$

$$\mathcal{N} = U(\pm 1)\mathcal{M}U(\mp 1) \,. \tag{48}$$

It is common to work with $(-)$-half-sided inclusions (and hence $(+)$-half-sided translations), and we follow this sign convention in the main text.

---

[27]There is a technical flaw in Wiesbrock's original proof, which was remedied in [124].

# B  Entropic musings

In this appendix, we collect some results on the behaviour of various entropy measures under the double-trace deformations discussed in the main text. We first discuss relative entropy, which furnishes the relationship between modular hamiltonians and entanglement entropy encoded in the eponymous "first law" [34]. We find that the expectation value of the generator (5) vanishes to first order, and admits an expression entirely in terms of the relative entropies between the associated algebras and their commutants. We then examine the entanglement entropy and mutual information between the left and right algebras, and find that these quantities appear to change in opposite directions in the bulk vs. the boundary. In the context of the emergent spacetime picture discussed in section 4, this calls for a more careful study of entanglement measures to diagnose connectivity of the interior spacetime.

## B.1  Modular hamiltonians and relative entropy

Recall the definition of relative entropy between two states $\rho_1$ and $\rho_0$:

$$S(\rho_1|\rho_0) = \text{tr}(\rho_1 \ln \rho_1) - \text{tr}(\rho_1 \ln \rho_0) = \delta \langle K_0 \rangle - \delta S \geq 0 \,, \tag{49}$$

where

$$\delta \langle K_0 \rangle = \text{tr}(\rho_1 K_0) - \text{tr}(\rho_0 K_0) \,, \qquad \text{and} \qquad \delta S = S(\rho_1) - S(\rho_0) \,, \tag{50}$$

and $K_0$ is the half-sided modular hamiltonian corresponding to the reduced density matrix $\rho_0$, i.e., $K_0 = -\ln \rho_0$. Note that we use $\delta$ to denote both finite and infinitesimal changes, so as to avoid confusion with the modular operator $\Delta$. Relative entropy has many important properties. For example, positivity of relative entropy was shown to underlie positivity of quantum mutual information in [126]. It is also monotonic under inclusions, i.e., $S(\rho_1^{\mathcal{M}}|\rho_0^{\mathcal{M}}) > S(\rho_1^{\mathcal{N}}|\rho_0^{\mathcal{N}})$ for $\mathcal{N} \subset \mathcal{M}$, where $\rho_1^{\mathcal{A}}$, $\rho_0^{\mathcal{A}}$ are taken to be an arbitrary state and the vacuum state, respectively, restricted to the region $\mathcal{A}$; this fact was used in [26] to demonstrate the positivity of the operator $p$ (7). Perhaps the most salient feature, at least in the holographic context, is the observation by [34] that the relative entropy vanishes for linear perturbations around the reference state $\rho_0$, and hence to first order,

$$\delta \langle K_0 \rangle = \delta S \,. \tag{51}$$

This is the so-called first law of entanglement entropy, and provides perhaps the cleanest connection between the modular hamiltonian and familiar notions of entropy.

To that end, it is natural to ask whether the operator $p$ associated with the inclusion structure in the main text has a similarly simple interpretation. Observe that this is the difference in the *full* (that is, two-sided) bulk modular hamiltonians; working from the perspective of the right wedge, we would therefore write the generator (7) as

$$2\pi p^b = \widehat{K}_{\mathcal{M}_R}^b - \widehat{K}_{\mathcal{M}_N}^b = (K_{\mathcal{M}_R}^b - K_{\mathcal{M}_R'}^b) - (K_{\mathcal{N}_R}^b - K_{\mathcal{N}_R'}^b) \,, \tag{52}$$

where here we denote the full-sided hamiltonians with hats, to distinguish them from their one-sided constituents without. To minimize clutter, we shall henceforth suppress the superscript "$b$", with the understanding that we're working from the perspective of the right bulk wedge.[28] The expectation value of (52) may then be written

$$2\pi \langle p \rangle = \langle K_{\mathcal{M}_R} - K_{\mathcal{N}_R} \rangle + \langle K_{\mathcal{N}_R'} - K_{\mathcal{M}_R'} \rangle = \langle \delta K \rangle + \langle \delta K' \rangle \,, \tag{53}$$

---

[28]Since the bulk and boundary relative entropies are equal, this distinction is not relevant anyway [32].

where we have defined

$$\delta K \equiv K_{\mathcal{M}_R} - K_{\mathcal{N}_R} , \qquad\qquad \delta K' \equiv K_{\mathcal{N}'_R} - K_{\mathcal{M}'_R} . \tag{54}$$

For our purposes, it is sufficient to work with $\langle \delta K \rangle$; the analysis for the commutant operator is precisely identical. Of course, evaluating this expectation value requires that we specify a state—two states, in fact, if we wish to connect with relative entropy. Let us therefore consider the change in the expectation value between the initial and final state, i.e.,

$$\delta \langle \delta K \rangle = \mathrm{tr}(\rho_{\mathcal{M}_R} \delta K) - \mathrm{tr}(\rho_{\mathcal{N}_R} \delta K) = -S(\mathcal{M}_R | \mathcal{N}_R) - S(\mathcal{N}_R | \mathcal{M}_R) , \tag{55}$$

where

$$
\begin{aligned}
S(\mathcal{M}_R | \mathcal{N}_R) &= -\mathrm{tr}(\rho_{\mathcal{M}_R} K_{\mathcal{M}_R}) + \mathrm{tr}(\rho_{\mathcal{M}_R} K_{\mathcal{N}_R}) , \\
S(\mathcal{N}_R | \mathcal{M}_R) &= -\mathrm{tr}(\rho_{\mathcal{N}_R} K_{\mathcal{N}_R}) + \mathrm{tr}(\rho_{\mathcal{N}_R} K_{\mathcal{M}_R}) .
\end{aligned}
\tag{56}
$$

Repeating this for $\delta K'$, we therefore find

$$2\pi \langle p \rangle = S(\mathcal{M}'_R | \mathcal{N}'_R) - S(\mathcal{M}_R | \mathcal{N}_R) + S(\mathcal{N}'_R | \mathcal{M}'_R) - S(\mathcal{N}_R | \mathcal{M}_R) . \tag{57}$$

Given the quadratic nature of relative entropy under perturbations mentioned above, this implies that $\langle p \rangle$ vanishes to first order for the case where $\mathcal{M}$ represents only a small deviation from $\mathcal{N}$. In some sense, this is disappointing, as it necessitates working to at least quadratic order to derive further insight; see for example [34] for an analysis of such second-order corrections. On the other hand, the ability to express $\langle p \rangle$ in terms of relative entropy makes the freedom from UV divergences manifest. That is, while the half-sided modular hamiltonians (and for that matter, the associated reduced density matrices) are not well-defined in type III theories, the full modular hamiltonian certainly is. Hence we would not expect this quantity to be directly expressible in terms of the (UV-divergent) von Neumann entropy. While in this sense, the expectation value of the generator for the modular inclusion $\mathcal{N} \subset \mathcal{M}$ provides a finite measure of distinguishability between the associated states, it would be interesting to explore whether more tractable or deeper relations exist.

## B.2 Entanglement and mutual information

It is important to distinguish between the boundary and bulk entropies/modular hamiltonians, as these contribute at different orders in the $1/N$ expansion.[29] Accordingly, we shall denote a boundary entity by $X^{\partial b}$, and the corresponding bulk quantity by $X^b$; unless otherwise indicated, the absence of either superscript indicates that a given expression holds on both sides. In particular, the bulk/boundary entropies [13] and modular hamiltonians [32] are related by

$$S^{\partial b} = \frac{\mathrm{Area}}{4 G_N} + S^b + \dots , \qquad\qquad K^{\partial b} = \frac{\mathrm{Area}}{4 G_N} + K^b + \dots , \tag{58}$$

where the leading-order area term is computed via the Ryu-Takayanagi prescription [8,9], and the ellipses denote higher-order terms which we shall ignore. Of course, entropy is a property of a state, and hence evaluating these expressions requires the selection of a particular Cauchy slice. Since we're interested in examining the effect of the deformation, we shall consider two constant time-slices through the bulk. The first, on which quantities will be denoted $X_{\mathcal{N}}$, is taken to join the two unperturbed CFTs at $t_L = t_R = 0$, such that it runs through the bifurcation point at the lower tip of the double-cone in fig. 1. The second, on which quantities will be denoted $X_{\mathcal{M}}$, is chosen to horizontally bisect this double-cone through the left- and right-tips (though in fact, our analysis holds for any such time-slice which crosses this overlap region.

---

[29]I am grateful to Michal Heller for emphasizing this point.

Since bulk and boundary states are dual, we shall suppress the corresponding superscript on the reduced density matrices; thus, for example, $\rho_{\mathcal{N}_R}$ will be taken to represent both the bulk state in the right wedge on the prescribed Cauchy slice, as well as the thermal density matrix for the right CFT at inverse temperature $\beta_{\mathcal{N}}$.

Now, let us begin by considering the change in the boundary entropy under the deformation,

$$\delta S^{\partial b} = S^{\partial b}_{\mathcal{M}_R} - S^{\partial b}_{\mathcal{N}_R} \; . \tag{59}$$

By (58), this expression consists of two parts, namely the change in the Ryu-Takayanagi area term and the change in the bulk entropy. Since $\mathcal{M}_R$, $\mathcal{N}_R$ are taken to have support on the entire (right) boundary, the Ryu-Takayanagi surface wraps the horizon of the black hole, so the difference in areas corresponds to the change in the entropy of the black hole, denoted $S^{\text{BH}}$. As illustrated in sec. 2, the size of the black hole decreases due to the negative-energy shockwave, and hence the contribution from the area term is negative,

$$\frac{\delta \text{Area}}{4 G_N} = \delta S^{\text{BH}} = S^{\text{BH}}_{\mathcal{M}_R} - S^{\text{BH}}_{\mathcal{N}_R} < 0 \; . \tag{60}$$

In contrast, the change in the bulk entropy is positive due to the enlargement of the exterior algebras,

$$\delta S^b = S^b_{\mathcal{M}_R} - S^b_{\mathcal{N}_R} > 0 \; . \tag{61}$$

Here it is important to bear in mind that the exterior algebra is given by $\mathcal{M}_{L,R}$, not the commutants $\mathcal{M}'_{L,R}$, otherwise the Penrose diagrams in the main text may lead to confusion. Recall that for any pure state, the entropy of a given region $A$ must be equal to the entropy of the complement $\bar{A}$, and furthermore that this is bounded by the smaller of the two regions. This is easy to see by imagining pairwise entangled degrees of freedom in $A$ and $\bar{A}$, in which case one will run out of points in the smaller region before the larger. Therefore, if the total system size were fixed, the fact that $\mathcal{M}'_R \subset \mathcal{N}'_R \cong \mathcal{N}_L \cong \mathcal{N}_R \subset \mathcal{M}_R$ (where $\cong$ denotes an isomorphism) would imply a smaller entanglement entropy between $\mathcal{M}_R$ and $\mathcal{M}'_R$ than between $\mathcal{N}_R$ and $\mathcal{N}'_R$. This however is not the relevant quantity, since the left algebra is given by $\mathcal{M}_L$, not $\mathcal{M}'_R$. In other words, the above reasoning fails because the total system size (i.e., the total exterior algebra) has increased: both $A$ and the formerly-complementary region $\bar{A}$ have grown, but in such a manner as to overlap. What is now $\bar{A}$ then occupies a smaller relative volume; i.e., the set of operators which commutes with the exterior right algebra has shrunk.

However, since the bulk entropy term is higher-order in the $1/N$ expansion, the change in the area term (60) dominates over (61), and hence the change in the boundary entropy (59) is negative,

$$\delta S^{\partial b} = \delta S_{\text{BH}} + \delta S^b < 0 \; . \tag{62}$$

Our interpretation of this result is as follows: initially, tracing over the left CFT resulted in an exactly thermal state, i.e.,

$$S^{\partial b}_{\mathcal{N}_R} = -\text{tr}(\rho_{\mathcal{N}_R} \ln \rho_{\mathcal{N}_R}) \;, \qquad \text{with} \qquad \rho_{\mathcal{N}_R} = \text{tr}_{\mathcal{N}'_R} \rho = Z^{-1}_{\beta_{\mathcal{N}}} e^{-\beta_{\mathcal{N}} H} \;, \tag{63}$$

where, on the boundary, $\rho = |TFD\rangle \langle TFD|$. After the deformation however, the bulk overlap $\mathcal{M}_R \cap \mathcal{M}_L = \mathcal{C}$ implies that we trace out less than half the total operator content of the boundary, i.e.,

$$S^{\partial b}_{\mathcal{M}_R} = -\text{tr}(\rho_{\mathcal{M}_R} \ln \rho_{\mathcal{M}_R}) \;, \qquad \text{with} \qquad \rho_{\mathcal{M}_R} = \text{tr}_{\mathcal{M}'_R} \rho \;. \tag{64}$$

The density matrix $\rho_{\mathcal{M}_R}$ is therefore not quite thermal, so the entropy $S^{\partial b}$ has decreased.

To make contact with the discussion in sec. 4, let us consider the change in the mutual information between the two sides. Since $\mathcal{M}_L \cap \mathcal{M}_R \neq \emptyset$, this is most cleanly quantified by the

information between the commutants, which are disjoint for both $\mathcal{N}$ and $\mathcal{M}$. Hence, working in the bulk, we have

$$\delta I^b = I^b(\mathcal{M}'_R, \mathcal{M}'_L) - I^b(\mathcal{N}'_R, \mathcal{N}'_L) \,, \tag{65}$$

where the mutual information between $A$ and $B$ is

$$I(A,B) = S_A + S_B - S_{A\cup B} \equiv S(\rho_A) + S(\rho_B) - S(\rho_{AB}) \,. \tag{66}$$

By symmetry, $\mathcal{N}_{L,R}$ is isomorphic to its commutant, so

$$I^b(\mathcal{N}'_R, \mathcal{N}'_L) = S^b_{\mathcal{N}_R} + S^b_{\mathcal{N}_L} - S^b_{\mathcal{N}_R \cup \mathcal{N}_L} = 2S^b_{\mathcal{N}_R} \,, \tag{67}$$

where the last step follows by left-right symmetry, and the fact that the total state is pure (i.e., $S_{\mathcal{N}_L \cup \mathcal{N}_R} = 0$). While this symmetry is broken for the $\mathcal{M}$ algebras, purity still enables us to identify the entropy of the algebra with its commutant, and hence

$$I^b(\mathcal{M}'_R, \mathcal{M}'_L) = S^b_{\mathcal{M}_R} + S^b_{\mathcal{M}_L} - S^b_{\mathcal{M}'_R \cup \mathcal{M}'_L} = 2S^b_{\mathcal{M}_R} - S_{\mathcal{C}} \,, \tag{68}$$

where the last step follows from the fact that tracing out $\mathcal{C}$ is equivalent – again by purity – to tracing out both $\mathcal{M}'_L$ and $\mathcal{M}'_R$. Substituting (68) and (67) into (65) then yields

$$\delta I^b = 2\,\delta S^b - S^b_{\mathcal{C}} \,, \tag{69}$$

via (61). Note that if we had instead examined the change in the mutual information between the algebra and its commutant, we would obtain simply

$$I^b(\mathcal{M}_R, \mathcal{M}'_R) - I^b(\mathcal{N}_R, \mathcal{N}'_R) = 2\,\delta S^b > 0 \,. \tag{70}$$

In other words, if we didn't create the overlapping region $\mathcal{C}$, we would conclude that the change in the bulk mutual information is simply due to the change in the entropy. Instead however, the change between the two sides, as measured by (69), is smaller due to the fact that the enlargement mentioned above goes into creating the region $\mathcal{C}$, whose information cannot be uniquely localized to either side. Of course, we can run the same story on the boundary, which leads to

$$\delta I^{\partial b} = 2\,\delta S^{\partial b} - S^{\partial b}_{\mathcal{C}} \,. \tag{71}$$

Since $S^{\partial b}_{\mathcal{C}}$ is positive, the decrease in boundary entropy (62) then implies that the boundary mutual information likewise decreases.

While one might have anticipated this last conclusion from considerations of scrambling – in particular, that the double-trace insertion disrupts the fine-grained correlations in the (highly atypical) TFD state – it is at first glance rather surprising, particularly in the context of the emergent spacetime picture in sec. 4. To see why, observe that we can apply the same analysis as above to the SS-type shocks which move the bulk exterior regions apart. In that case, we would find precisely the opposite of the above: namely, that the entropy and mutual information in the bulk decrease, but the corresponding quantities on the boundary increase—the latter from the simple fact that the boundary quantities are dominated by the leading-order area term, cf. (58), and the SS-type shocks increase the size of the black hole. The fact that the boundary mutual information moves in opposite directions from that suggested by fig. 3 thus seems at odds with both van Raamsdonk's heuristic picture of increasing separation under SS-type shocks [53] as well as the analysis in [88], who explicitly computed the decrease in mutual information between boundary regions.

However, the mutual information considered herein differs from that considered in the aforementioned works, both of which considered the mutual information between *subregions* $A$ and $B$ on either boundary. In contrast, we consider the mutual information between the

exterior algebras of CFT operators, whose spatial support covers the entire boundary. Furthermore, it is unclear whether these algebras contain the heavy operators whose two-point correlators exhibit the unambiguous exponential fall-off with geodesic distance through the bulk, which underlied the heuristic argument for increased spacetime separation in [53]. The second key difference is that [88] considered the increase in said mutual information under time evolution.[30] In this case, for sufficiently large regions, the term $S_{A \cup B}$ which joins the two regions through the wormhole contains a time-dependent contribution; for positive-energy shockwaves, this term eventually kills the mutual information after a time of order the scrambling time $t_*$. In contrast, taking the subsystems to comprise the entire boundary implies that this term is absent in our treatment, which furthermore regards the initial and final states as static geometries (one can consider evaluating the mutual information after the system has returned to equilibrium). Therefore, in the present case the difference is purely determined by the difference in the horizon areas, and in this sense $\delta I^{\partial b}$ is a less refined probe of bulk dynamics. Hence for both of these reasons – namely the exclusion of heavy states, and the fact that the mutual information only receives contributions from the (static) horizon area – we suggest that the bulk mutual information $\delta I^b$ may provide a better diagnostic of spacetime connectivity in this context, to wit: it is the mutual information between the algebras of exterior operators that yields ground to the formerly-interior spacetime region $\mathcal{C}$. More generally, we hope that the considerations herein will motivate a more careful treatment of these issues, in order to identify the precise criteria on the boundary that herald the emergence (or disentangling) of spacetime in the bulk.

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
