# Peer review of "Comments on black hole interiors and modular inclusions"

_SciPost Physics, doi:SciPost Phys. 6, 042 (2019)_

## Round 2 · Referee Report · Anonymous · 2019-2-23

Strengths

1. Use of concepts in algebraic quantum field theory together with physical arguments to discuss aspects of black hole interiors and related matters.
2. The manuscript attempts to be self-contained.
3. The main goals of the manuscript go through.

Weaknesses

1. Few statements may generate some confusions among the readers and some clarification may improve its readership.

Report

The work “Comments on black hole interiors and modular inclusions” is a timely and interesting piece of work providing some insightful remarks on the interpretation of black hole interiors using tools from algebraic quantum field theory.

Looking at the manuscript as a whole, I believe the two main points made in this work : i) that state dependence is inherited from the requirement of describing the physics of the interior of BHs by imposing operators to be unitary and local (using the exterior algebra) and ii) to give evidence for the relevance of modular inclusions to describe traversable wormholes by enlarging the notion of exterior algebra, in a process that is kind of opposite to the one making it smaller, i.e. the one appearing in shock-wave physics as in Shenker-Stanford, do go through.

Even though the main technical remarks and proofs are rather abstract (building on notions developed in algebraic quantum field theory), the main body of the text always provides with physical arguments and interesting digressions/comparisons with existent arguments in the literature regarding bulk reconstruction, the construction of mirror operators and their state dependence, relation to quantum error correction and the language of code subspaces. The manuscript also makes an attempt at being self-contained by including a pair of appendices in which the Tomika-Takesaki theory is briefly reviewed and implications for relative entropy and mutual information discussed from both bulk and boundary perspectives are discussed.

Referencing is good.

To sum up, even though this is a topic with many open questions left and part of the content in this manuscript is reviewing material first introduced in other references, I find the perspective offered by modular inclusions insightful and gives the author the chance to provide good physical arguments commenting on previous work that some readers will find inspiring and clarifying. For all these reasons, I recommend to publish this manuscript in this journal, though it would benefit its readership if the author could act upon the points raised above.

For completeness, let me mention some typos : there are several “analagous” through the entire text that should be replaced by “analogous”. Furthermore, I believe the word “found” in page 18 in the paragraph ending with the footnote 22 should be replaced by “find”.

Requested changes

1. In section 2, when referring to the teleological nature of event horizons, it is claimed the double trace deformation does not allow information inside the black hole to escape because it would violate causality. I am not sure what the author is alluding to here. Perhaps this is related to the use of quotation marks for the word inside. Is this a classical statement ? How does Hawking-radiation fit into this statement ? It would help some readers if the author can explain this point in a slightly different way.
2. In section 2, and currently at the start of page 6, the notion of “exterior operators” is used. Though eventually there should be no doubt on what the author is referring to, it may help to define more precisely what they mean (operators belonging to the von Neumann algebras just introduced).
3. In section 2, I would suggest to break the long paragraph at the start of page 6. In particular, the sentence starting with “Lastly” where new notation is being introduced could easily be moved to a separate paragraph. In the current version, one is expecting a further argument to support the discussion one is reading about, but encounters some further notation ending with a definition.
4. In section 2, by the end of the first paragraph in page 8, I find the statement “more low-energy operators are required to effect the same back reaction” confusing. It seems to me the author wants to say more low-energy operators are required to encode the information of a larger amount of exterior spacetime, which makes perfect sense with the arguments presented. I would suggest to improve this sentence to prevent any confusions among the readers.

---

## Round 3 · Referee Report · Anonymous · 2019-3-20

Report

I would like to thank the author for the time taken to properly address the small list of requested changes. I do recommend publication in its current format.

---

## Round 3 · Author Response

I would first like to thank the referee for their careful reading of the manuscript. Their report reflects precisely the key points I wished to convey, as well as the broader aim of the work, and I am delighted they found it of interest. The referee has made some helpful suggestions for improving the clarity of the paper, particularly by highlighting a few statements which could generate confusion; I have responded to these requested changes in detail in the "List of Changes" accompanying this resubmission. I am also grateful to the referee for catching the misspelling of "analogous". The instance of "found" in the third paragraph of section 5 is however correct as written (it is used here in the sense of "to ground, base, or establish").

I believe these changes have improved the manuscript, and thank the referee again for their detailed report.

Incidentally, as a matter of professional policy, I would like to politely object to the use of the masculine pronoun in requested change #2. While I do not believe the referee intended to misgender me, it is precisely for this reason that the gender of the author(s) should never be assumed. Rather, gender-neutral terms such as "the author" or the singular "they" should be used instead. The referee rightly does this throughout the rest of the report, but I believe this is nonetheless important to mention in the interest of cultivating a more inclusive scientific culture.

---

## Round 3 · List of Changes

In response to the referee's requested changes:

1. This is not a classical vs. quantum statement, but merely an attempt to emphasize the global definition of event horizons---namely, that contrary to existing depictions in the literature, the deformation does not cause the observer to hop out of the black hole. I agree with the referee that simply putting "inside" in quotations marks was ambiguous, particularly in light of the firewall debate, and have slightly modified both instances thereof on pages 4-5 in order to clarify my meaning.

2. While the meaning of exterior operators is discussed in more detail later in the manuscript, I have elaborated upon this first instance in an effort to mitigate any initial confusion.

3. I have broken the rather lengthy paragraph into two, as per the referee's sensible suggestion.

4. The referee is correct; I have modified the sentence to avoid such unintentionally cryptic phrasing.

---

## Editorial Decision

published